# Review on Active and Passive Remote Sensing Techniques for Road Extraction

**Jianxin Jia** [1], **Haibin Sun** [1,2], **Changhui Jiang** [1], **Kirsi Karila** [1], **Mika Karjalainen** [1], **Eero Ahokas** [1], **Ehsan Khoramshahi** [1], **Peilun Hu** [1,3], **Chen Chen** [1,4], **Tianru Xue** [1,2], **Tinghuai Wang** [5], **Yuwei Chen** [1,*] and **Juha Hyyppä** [1]

1    Department of Remote Sensing and Photogrammetry, Finnish Geospatial Research Institute,
     02430 Kirkkonummi, Finland; jianxin.jia@nls.fi (J.J.); sunhaibin007@gmail.com (H.S.);
     changhui.jiang@nls.fi (C.J.); kirsi.karila@nls.fi (K.K.); mika.karjalainen@nls.fi (M.K.); eero.ahokas@nls.fi (E.A.);
     ehsan.khoramshahi@nls.fi (E.K.); peilun.hu@helsinki.fi (P.H.); chenchen115039@njust.edu.cn (C.C.);
     xuetianru@mail.sitp.ac.cn (T.X.); juha.hyyppa@nls.fi (J.H.)
2    Key Laboratory of Intelligent Infrared Perception, Shanghai Institute of Technical Physics, Chinese Academy
     of Sciences, Shanghai 200083, China
3    Department of Forest Science, University of Helsinki, 00100 Helsinki, Finland
4    School of Automation, Nanjing University of Science and Technology, Nanjing 210094, China
5    Huawei Helsinki Research Centre, 00180 Helsinki, Finland; tinghuaiwang@huawei.com
*    Correspondence: yuwei.chen@nls.fi

**Abstract:** Digital maps of road networks are a vital part of digital cities and intelligent transportation. In this paper, we provide a comprehensive review on road extraction based on various remote sensing data sources, including high-resolution images, hyperspectral images, synthetic aperture radar images, and light detection and ranging. This review is divided into three parts. Part 1 provides an overview of the existing data acquisition techniques for road extraction, including data acquisition methods, typical sensors, application status, and prospects. Part 2 underlines the main road extraction methods based on four data sources. In this section, road extraction methods based on different data sources are described and analysed in detail. Part 3 presents the combined application of multisource data for road extraction. Evidently, different data acquisition techniques have unique advantages, and the combination of multiple sources can improve the accuracy of road extraction. The main aim of this review is to provide a comprehensive reference for research on existing road extraction technologies.

**Keywords:** road extraction; high-resolution image; hyperspectral image; synthetic aperture radar (SAR); light detection and ranging (LiDAR)

## 1. Introduction

Digital mapping of road networks is necessary for various industrial applications such as land use and land cover mapping [1], geographic information system updates [2,3] and natural disaster warning [4]. Moreover, it is a critical requirement for digital cities and intelligent transportation [5]. Traditional cartographic techniques are time-consuming and labour-intensive [6,7]. In comparison, remote sensing techniques changed the mapping community fundamentally without relying entirely on surveyed ground measurements [6]. Remote sensing data used for road extraction include ground moving target indicator (GMTI) tracking, smart phones global positioning system (GPS) data, street view images, synthetic aperture radar (SAR) images, light detection and ranging (LiDAR) data, high-resolution images, and hyperspectral images. GMIT radar has been used for extracting road map information due to the advantages of all-weather, real-time capabilities, and wide-area [8,9]. Smart phone GPS data were used for extracting road centrelines and monitoring road and traffic conditions [10,11]. Street view images obtained by Google of USA and Baidu of China companies have been used to detect, classify, and map traffic

signs and road crack information extraction [12,13]. In this paper, based on the different data sources, the existing road extraction technology can be roughly divided into four methods: high-resolution imaging-based, hyperspectral imaging-based, SAR imaging-based and LiDAR-based methods. SAR and LiDAR are active information acquisition methods. In contrast, high-resolution and hyperspectral imaging are passive optical imaging approaches. Each road extraction method, based on a different data source, has unique characteristics. For instance, the high-resolution imaging technology can be used to obtain images with centimetre-level accuracy and detailed target information [14]; hyperspectral remote sensing images are used for conventional road extraction; they also demonstrate excellent potential for road condition detection owing to a large number of bands (generally more than 100 bands) and continuous spectrum coverage [15]; SAR and LiDAR datasets are not easily affected by environmental factors such as changes in environmental illumination conditions or weather [16].

Road extraction is a popular research topic and has attracted the interest of numerous researchers. More than 2,770,000 results can be found in Google Scholar using the keyword 'road extraction', including some state-of-the-art reviews published in recent years. Wang et al. [17] summarised the main road extraction methods from 1984 to 2014 based on high-resolution images. In this study, road information extraction methods were divided into knowledge-based [18,19], classification-based [20–26], active contour-based [27–29], mathematical morphology-based [30,31], and dynamic programming-based grouping methods [32,33]. However, these methods were chiefly heuristic, and deep learning-based approaches were not presented [34]. This scenario has undergone a drastic change in recent years with the rapid development of patch-based convolutional neural networks (CNNs) [35–40], full convolutional network (FCN)-based [41–46], deconvolutional net-based [47–64], generative adversarial network (GAN)-based [41,65,66], and graph-based deep learning methods [67–70] for road extraction. Road extraction methods based on deep learning are collectively referred to as data-driven approaches in [34,71]. Abdollahi et al. [34] and Lian et al. [71] presented and compared the deep learning-based state-of-the-art road extraction methods using publicly available high-resolution image datasets. Sun et al. [72] reviewed the SAR image-based road extraction method. This review initially introduces the road characteristics and basic strategies and then presents a summary of the main road extraction techniques based on SAR images. Similarly, Sun et al. [73] analysed and summarised the SAR image-based road segmentation methods. They introduced the traditional edge detection and deep learning-based road segmentation methods and predicted that new segmentation methods of deep neural networks based on the self-attention mechanism and capsule paradigm as the future development trends. Wang and Weng [74] summarised the road extraction techniques based on LiDAR. In this report, road clusters were defined using the classification framework and algorithms for LiDAR point data-based road identification. Furthermore, techniques for generating road networks, including road classification refinement and centreline extraction, were summarised as well. Several other similar reviews [75–77] that provide scientific references for road extraction have been reported in recent years.

To the best of our understanding, the existing reviews on road extraction methods are commonly based on only a single data source; hence, they fail to provide a comprehensive view that can be derived using different data sources. However, a comprehensive road extraction review based on high-resolution imaging, hyperspectral imaging, SAR imaging, and LiDAR technologies is crucial to bridge the gap between potential applications and available technologies of road extraction. Thus, the aim of this study was to achieve this goal by combining the road extraction techniques based on diverse data sources, including high-resolution images, hyperspectral images, SAR images, and LiDAR data. In Section 2, we provide an overview of the four techniques and summarise the typically used sensors. In Section 3, we introduce and analyse the main road extraction methods using various data sources as well as summarise the road extraction status and prospects for different data sources. Finally, different combinations of the road extraction techniques are presented

in Section 4. To the best of our knowledge, in this paper, we present the first comprehensive review of road extraction, including high-resolution images, hyperspectral images, SAR images and LiDAR data sources.

## 2. Overview of the Existing Data Acquisition Techniques for Road Extraction

### 2.1. High-Resolution Imaging Technology

In this review, high-resolution images refer to high-spatial-resolution images (resolution of less than 10 m) that were mainly acquired using airborne or spaceborne sensors. The spatial resolution of the images refers to the size of a single pixel. High-resolution images are usually divided into two categories: panchromatic and multispectral images [78].

### 2.1.1. Data Acquisition Methods and Characteristics

High-resolution images are primarily recorded using spaceborne and airborne sensors. Spaceborne high-resolution imaging techniques have a broad area coverage and stable revisit periods; however, the cost of the satellite is high, and the images are easily affected by the atmosphere [79]. Compared to the spaceborne high-resolution images, the airborne high-resolution images possess higher resolutions and are less affected by the atmosphere. However, the working efficiency of airborne cameras is lower than that of the spaceborne instruments because of the lower flight altitude and smaller coverage [80]. Airborne high-resolution images can be obtained using manned aircrafts and unmanned aerial vehicles (UAVs). In recent years, several imaging systems with high-resolution cameras mounted on UAVs have been rapidly developed; these systems can achieve centimetre-level spatial resolution [81].

### 2.1.2. Typical Sensors

Spaceborne high-resolution imaging is still the main technical approach for earth observation. As shown in Table 1, an increasing number of high-resolution satellites have been developed; some of these satellites have been developed in series and are constantly being upgraded [82–86]. It can be seen from Table 1 that most high-resolution satellites were developed and launched by the USA, while the number of high-resolution satellites in China has increased in recent years. With the development of related technologies, the performance of spaceborne cameras continues to improve, and images with a spatial resolution better than 1 m can now be obtained.

**Table 1.** Main parameters of typical high-resolution satellites.

| Satellite | Launch (Year) | Swath (km) | PAN (m) | R (m) | G (m) | B (m) | NIR (m) |
|---|---|---|---|---|---|---|---|
| Gaofen 1 (CN) | 2013 [87] | | 2 | 8 | 8 | 8 | 8 |
| Gaofen 2 (CN) | 2014 [88] | 70 | 0.8 | 3.2 | 3.2 | 3.2 | 3.2 |
| Gaofen 6 (CN) | 2015 [89] | | 2 | 8 | 8 | 8 | 8 |
| SuperView (CN) | 2016 [90] | 12 | 0.5 | 2 | 2 | 2 | 2 |
| GeoEye 1 (US) | 2008 [91] | 15.2 | 0.41 | 1.65 | 1.65 | 1.65 | 1.65 |
| IKONOS (US) | 1999 [85] | 11.3 | 1 | 4 | 4 | 4 | 4 |
| PlanetScope (US) | 2018 [92] | 24.6 | / | 3 | 3 | 3 | 3 |
| QuickBirds (US) | 2001 [93] | 16.5 | 0.6 | 2.4 | 2.4 | 2.4 | 2.4 |
| WorldView 1 (US) | 2007 [94] | 17 | 0.5 | / | / | / | / |
| WorldView 2 (US) | 2009 [91] | 17 | 0.5 | 2 | 2 | 2 | 2 |
| WorldView 3 (US) | 2014 [95] | 13.1 | 0.31 | 1.24 | 1.24 | 1.24 | 1.24 |
| WorldView 4 (US) | 2016 [96] | 13.1 | 0.31 | 1.24 | 1.24 | 1.24 | 1.24 |
| OrbView 3 (US) | 2003 [97] | 8 | 1 | 4 | 4 | 4 | 4 |
| RapidEye (DE) | 2008 [98] | 77 | / | 6.5 | 6.5 | 6.5 | 6.5 |
| KOMPSAT 2 (KR) | 2006 [99] | 15 | 1 | 4 | 4 | 4 | 4 |
| KOMPSAT 3 (KR) | 2012 [100] | 16 | 0.7 | 2.8 | 2.8 | 2.8 | 2.8 |
| KOMPSAT 3A (KR) | 2015 [101] | 12 | 0.55 | 2.2 | 2.2 | 2.2 | 2.2 |
| Pléiades 1A (FR) | 2011 [102] | 20 | 0.7 | 2.8 | 2.8 | 2.8 | 2.8 |
| Pléiades 1B (FR) | 2012 [103] | 20 | 0.7 | 2.8 | 2.8 | 2.8 | 2.8 |
| SPOT 6 (FR) | 2012 [104] | 60 | 1.5 | 6 | 6 | 6 | 6 |
| SPOT 7 (FR) | 2014 [105] | 60 | 1.5 | 6 | 6 | 6 | 6 |
| DubaiSat 1 (AE) | 2009 [106] | 12 | 2.5 | 5 | 5 | 5 | 5 |
| DubaiSat 2 (AE) | 2013 [107] | 12 | 1 | 4 | 4 | 4 | 4 |

PAN: panchromatic. R: red. G: green. B: blue. NIR: near-infrared. WorldView 2 [91] and WorldView 3 [95] have four other multispectral bands (red edge, coastal, yellow and NIR), and RapidEye has another multispectral band (red edge) [98].

### 2.1.3. Application Status and Prospects

High-resolution images usually contain feature-rich information such as spectral characteristics, geometric features, and texture features, and hence, a significant amount of useful information can be extracted from such images. High-resolution imaging has been widely used in forest management [108], urban mapping [109], farmland management [110], disaster and security mapping, public information service and environmental monitoring. Numerous high-resolution satellites have been developed and launched in recent years. These satellites can form satellite networks to obtain image data with a wide coverage. However, the huge amounts of data also bring new challenges to data transmission and processing [80]. Such high-resolution images have been extensively used for road extraction [34,71]. Moreover, several commercial products such as Google Maps based on high-resolution images have been successfully developed and applied in many fields in recent years.

### 2.2. Hyperspectral Imaging Technology

Hyperspectral imaging technology is another commonly used technique for obtaining the spectra of a target [111,112]. The hyperspectral image containing two-dimensional (2D) spatial and 1D spectral information comprises a 3D data cube [113]. Notably, different objects exhibit different spectra, which can be used for the identification and detection of such objects. In the 1980s, Goetz et al. [114] began a revolution in remote sensing by developing an airborne visible infrared imaging spectrometer (AVIRIS) [113], which initiated the development of hyperspectral imagers. The number of bands in a multispectral image is usually less than five, while that in a hyperspectral image is more than 100; moreover, continuous spectral information is obtained from a hyperspectral image.

### 2.2.1. Data Acquisition Methods and Characteristics

Hyperspectral images are obtained by an imaging spectrometer, which is a complex and sophisticated optical system that includes several subsystems and components. The main components of the sensor are a scan mirror, fore-optics, spectrometers, detectors, onboard calibrators and electronic units. The fore-optics of the system receive light, which is dispersed by a spectrometer and converted from photons to electrons by the detector to yield an electronic signal. This electronic signal is then amplified, digitised and recorded by the electronic unit. The instrument performance and preprocessing data results are the main factors that aid in acquiring high-accuracy surface reflectance data. Such an instrument is characterised by its field-of-view (FOV), spectral range, spatial and spectral resolution and sensitivity. Data preprocessing includes geometric rectification, calibration and atmospheric correction [80].

### 2.2.2. Typical Sensors

According to the installed platforms, hyperspectral sensors can be divided into spaceborne, airborne, UAV [115–117], car-borne [118], and ground-based [119] imaging systems. Airborne hyperspectral imaging systems were the first hyperspectral imagers to be developed and used for verifying the design of later spaceborne instruments. Jia et al. [80] presented a comprehensive review of airborne hyperspectral imagers, including key design technologies, preprocessing and new applications. Based on this review, we added spaceborne sensors in this paper and summarised the typical hyperspectral imagers in Table 2. Evidently, most hyperspectral sensors, especially the spaceborne hyperspectral imagers, were developed in the USA. In addition, there are more airborne hyperspectral imagers are than spaceborne hyperspectral imagers due to the hardware investment. The future spaceborne program includes the HyspIRI hyperspectral satellite of America [120] and the ENMAP hyperspectral satellite of Germany [121]. Most hyperspectral imagers can acquire data in the visible to near-infrared spectral range owing to the availability of silicon detectors with wide spectral detection ranges. Additionally, shortwave infrared and longwave infrared hyperspectral imagers have emerged in recent years [80].

**Table 2.** Typical airborne and spaceborne hyperspectral sensors.

| Name | References | Platform | Wavelength Range (μm) | Channel | Spectral Resolution (nm) | IFOV (mrad) | FOV/Swath |
|---|---|---|---|---|---|---|---|
| AISA-FENIX 1K | [122], 2018 | Airborne | 0.38–0.97, 0.97–2.5 | 348, 246 | $\leq$4.5, $\leq$12 | 0.68 | 40° |
| APEX | [123], 2015 | Airborne | 0.372–1.015 0.94–2.54 | 114, 198 | 0.45–0.75, 5–10 | 0.489 | 28.1° |
| AVIRIS-NG | [124,125], 2016, 2017 | Airborne | 0.38–2.52 | 430 | 5 | 1 | 34° |
| CASI-1500 SASI-1000A TASI-600A | [126], 2014 | Airborne | 0.38–1.05, 0.95–2.45, 8–11.5 | 288, 100,32 | 2.3, 15, 110 | 0.49, 1.22, 1.19 | 40° |
| AMMIS | [127,128], 2019, 2020 | Airborne | 0.4–0.95, 0.95–2.5, 8–12.5 | 256, 512, 128 | 2.34, 3, 32 | 0.25, 0.5, 1 | 40° |
| SYSIPHE | [129], 2016 | Airborne | 0.4–1, 0.95–2.5, 3–5.4, 8.1–11.8 | 560 (total) | 5, 6.1, 11 cm$^{-1}$, 5 cm$^{-1}$ | 0.25 | 15° |
| HSI | [130], 1996 | LEWIS Satellite | 0.4–1, 1–2.5 | 128, 256 | 5, 5.8 | 0.057 | 7.68 km |
| Hyperion | [131], 2003 | EO-1 Satellite | 0.4–1, 0.9–2.5 | 242 (total) | 10 | 0.043 | 7.7 km |
| CHRIS | [132], 2004 | PROBA-1 Satellite | 0.4–1.05 | 18/62 | 1.25–11 | 0.03 | 18.6 km |
| CRISM | [133], 2007 | MRO Satellite | 0.362–1.053, 1.002–3.92 | 544 (total) | 6.55 | 0.061 | >7.5 km |
| AHSI | [134], 2019 | Gaofen-5 Satellite | 0.39–2.51 | 330 (total) | 5, 10 | 0.043 | 60 km |

IFOV: instantaneous field of view.

### 2.2.3. Application Status and Prospects

Hyperspectral imaging—a quantitative remote sensing approach—has been widely applied in environmental monitoring, vegetation analysis, geologic mapping, atmospheric characterisation, biological detection, camouflage detection and disaster assessment [135–141]. However, the application requirements for hyperspectral sensors underwent significant variations with the development of advanced sensors. First, a wide spectrum covering range is required to enhance the monitoring and detection capabilities of this system in various applications. This can be achieved by combining multiple sensors with different wavelength detection ranges [142] or by using an integrated system with a wide spectral range [80]. Second, the system sensitivity and preprocessing accuracies are equally important along with the spatial and spectral resolutions. For example, the AVIRIS next generation system [143] has been applied to detect methane, owing to its high signal-to-noise ratio and high data preprocessing accuracy. Finally, this technology facilitates the advantages of characterisation and quantification of targets; for example, hyperspectral imagers have been used for road network extraction.

### 2.3. SAR Imaging Technology

2.3.1. Data Acquisition Methods and Characteristics

SAR is an active remote sensing technology that uses microwaves with wavelengths of few centimetres as opposed to LiDAR, which functions using optical wavelengths (ultraviolet, visible, near-infrared or shortwave infrared light). Both these sensors measure the distance between the instrument and the target using the time delay of the echoes.

One of the main benefits of SAR is the acquisition of fine and detailed images through the clouds; moreover, this sensor can even work at night. In SAR, initially, short-pulsed microwave radiation is emitted and backscattered by the target; this backscattered signal from the illuminated area is then recorded. A SAR system with a long virtual antenna can generate fine spatial resolution. Among the currently available SAR systems, the spaceborne SAR sensors can provide sub-metre spatial resolutions. Notably, the long

antenna aperture is realised in the cross-range direction and that the range resolution is given by the pulse width (in general, the bandwidth of the signal). Another particularity of the SAR compared with optical sensors is the low correlation between range and spatial resolution.

SAR uses a side-looking imaging geometry to generate 2D image data of the target area. In target areas with uneven topography, the side-looking imaging geometry distorts the SAR images because of various factors such as foreshortening, layover and radar shadows [144]. These distortions are challenging for mapping applications, especially in urban areas with high-rise buildings.

SAR sensors collect data in a complex domain, and the acquired data can be converted into intensity and phase information. SAR data are usually presented as 2D intensity images that provide information on the amount of backscattered signal. Since the backscattered signal is a combination of signals from multiple scatterers, the images have granular noise called speckle. Sensors transmit and receive horizontally and vertically polarised signals and provide either single, dual or quad polarisation data [145–147]. Phase information is used in SAR polarimetry and interferometry. Interferometric coherence, that is, the complex correlation coefficient between two SAR images, can provide information on the changes in the target changes and can be used in target classification as well [148–151].

### 2.3.2. Typical Sensors

A list of typical SAR satellite systems was recently presented in [152]. With the spotlight imaging mode, less than 1 m spatial resolutions can be achieved (e.g., RCM, Cosmo-Skymed, Terrasar-X, ICEYE). However, the area covered by the image is limited. In general, the swath width varies from 5 to 500 km, and for a wide swath, the resolution is of the order of tens of meters. Spaceborne systems use the X-, C-, S-, L- or P-band sensors. X-band sensors provide the highest resolution; however, their penetration into the vegetation canopy is limited [153]. Furthermore, the polarimetric capabilities of satellites vary significantly; for example, Alos-2, Radarsat-2 and Terrasar-X are fully polarimetric, providing HH, VV, VH and HV data (where HH is horizontal transmit, horizontal receive; VV is vertical transmit, vertical receive; VH is vertical transmit, horizontal receive and HV is horizontal transmit, vertical receive); Cosmo-Skymed and Sentinel-1 provide dual polarimetric HH and VV data and ICEYE provides single polarisation (VV) data. The incidence angle in most satellites can be adjusted between 10° and 60°.

The data recorded by Sentinel-1 of the European Copernicus system are openly and freely accessible. The Sentinel-1 data is similar to those of the previous European Envisat SAR; however, the data availability has increased because of the use of multiple satellites.

When satellites fly in a constellation, short revisit times are possible. Cosmo-Skymed (four satellites) and SAR-Lupe (five satellites) constellations were designed to provide intelligence information. New commercial microsatellite constellations such as ICEYE (10/18 satellites launched), Spacety (18/56 launched) and Capella XSAR (18/36 launched), can provide good temporal coverage. A particular constellation is also formed by TerraSAR-X and Tandem-X, allowing single-pass interferometry. Based on the data collected by these satellite constellations, a global digital elevation model (DEM) has been developed [154].

Airborne SAR systems that enable data acquisition at different wavelengths, higher resolutions and single-pass interferometry are more versatile than the spaceborne systems. Most of these airborne systems are used for research purposes; for instance, the German Aerospace Centre (DLR) has an F-SAR [155] system operating on a Dornier 228 aircraft, providing fully polarimetric data in the X-, C-, S-, L- and P-bands (maximum four bands simultaneously); this system enables single-pass interferometry in the X- and S-bands. In addition, several UAV SAR sensors are also available in the market; for example, SAR Aero offers 1.8 kg SAR sensors in the X-or L-band with 0.3 to 3 m for a range of up to 10 km.

### 2.3.3. Application Status and Prospects

The principal benefit of SAR is its all-day and all-weather imaging capability, enabling rapid mapping [156]. Therefore, the main application areas are related to emergency and security-related services, where the timeliness and availability of data are critical; for example, SAR data are operationally used in sea-ice mapping, which cannot be easily extracted using other remote sensing techniques, especially in cloudy winters. In addition, considerable scientific research has been conducted on agricultural monitoring, forest mapping and topographic mapping. Continuous environmental monitoring is possible using spaceborne SAR datasets. Moreover, some previously reported studies utilised SAR images for road extraction [72,157,158].

### 2.4. Airborne Laser Scanning (ALS)

### 2.4.1. Data Acquisition Methods and Characteristics

In airborne laser scanning (ALS), a LiDAR sensor is mounted on an aircraft, along with an inertial measurement unit and a global navigation satellite system (GNSS) receiver. The LiDAR sensor transmits narrow laser pulses towards the ground and generates a scanning pattern over the target area. ALS systems, typically based on an oscillating mirror and scanning patterns, receive the return signal, measure the time of signal travel and associate each return pulse with the GNSS time and scan angle at which the pulse was transmitted. The travel time can be converted to distance and then to height. The ALS technique can produce georeferenced 3D point clouds in the target area [75,159,160].

The operational ALS systems are mostly based on single-wavelength single-pulse linear-mode LiDAR. The new emerging multispectral ALS systems use a combination of LiDARs at different wavelengths. These sensors provide intensity data that can be used to derive colour images, such as optical imagery. The chief advantage of this technique is that the acquired data are independent of illumination conditions and are without shadows. Therefore, multispectral ALS systems have great potential for increasing the automation level in mapping. Geiger-mode LiDAR and single-photon LiDAR (SPL) are new ALS techniques that are sensitive to a single photon and can provide dense point clouds from higher flight altitudes, owing to their higher system sensitivity.

### 2.4.2. Typical Sensors

The biggest ALS manufacturers include Leica Geosystems (Switzerland), Teledyne Optech (Canada) and RIEGL (Austria). Examples of the current system are listed in Table 3. It can be seen from Table 3 that ALS can collect one to several million points per second, which secure the usability of the collected data for most surveyed and mapping cases. Meanwhile, the lidar systems can be used both the low-altitude platforms (UAV and helicopter) and high-altitude ones (fixed-wing aircraft). In addition, most ALSs do not operate in eye-safety wavelength: typical operating wavelengths for ALS systems are 532 (green), 1064 (near-infrared) and 1550 nm (shortwave infrared). The point density and accuracy depend on the flying height, reaching a maximum of 60 points/m$^2$. In addition, its accuracy depends on the range measurement accuracy combining with attitude measurement accuracy. In addition, small sensors are available for UAVs.

**Table 3.** Examples of currently available commercial ALS systems.

| | Special Characteristics | WaveLength | Horizontal and Elevation Accuracy | Altitude | Pulse Repetition Frequency | Point Density |
|---|---|---|---|---|---|---|
| Leica Hyperion2+ [161], 2021 | Multiple pulses in the air measured | 1064 nm | <13 cm, <5 cm | 300–5500 m | −2000 kHz | 2 pts/m$^2$/4000 m, 40 pts/m$^2$/600 m |
| Leica SPL [162], 2021 | Single photon | 532 nm | <15 cm, <10 cm | 2000–4500 m | 20–60 kHz | 6 million points per second, 20 pts/m$^2$ (4000 m AGL) |
| Optech Galaxy Prime [163], 2020 | Wide-area mapping | 1064 nm | 1/10,000 × altitude, <0.03–0.25 m | 150–6000 m | 10–1000 kHz | 1 million point per s, 60 pts/m$^2$ (500 m AGL), 2 pts/m$^2$ (3000 m) |
| Optech Titan [164], 2015 | 3 wavelength | 1550 nm, 1064 nm, 532 nm | 1/7500 × altitude, <5–10 cm | 300–2000 m | 3 × 50–300 kHz | 45 pts/m$^2$ (400 AGL) |
| Riegl VQ-1560i-DW [165], 2019 | Dual-wavelength, multiple pulses in the air measured. | 532 nm, 1064 nm | / | 900–2500 m | 2 × 700–1000 kHz | 2 × 666,000 pts/s, 20 pts/m$^2$ (1000 m AGL) |
| Riegl Vux-240 [166], 2021 | UAV | 1550 nm | <0.05 m <0.1 m | 250–1400 m | 150–1800 kHz | 60 pts/m$^2$ (300 m) |

### 2.4.3. Application Status and Prospects

ALS produces accurate 3D models of the target areas. The main application areas of ALS are in topographic mapping, particularly DEM production, city modelling and forestry. Even though the area covered by the ALS in a single scan is limited compared to that covered by the spaceborne sensors, nationwide datasets are available, especially for the Nordic countries. In addition, ALS is currently used to detect human activity in archaeology [167].

## 3. Road Extraction Based on Different Data Sources

High-resolution images, hyperspectral images, SAR images and LiDAR data are primarily used for road extraction. To date, various road extraction methods have been presented in previously reported studies, and the observed differences are due to the use of different data sources. In this section, we summarise the methods, application status and prospects of road extraction based on four data sources.

### 3.1. Road Extraction Based on High-Spatial Resolution Images

Extracting road information from high-resolution images requires clarification of the road features, including the radiation features, geometric features, topological features and texture features [71]. First, information on the various elements such as features, textures and edges are extracted from the image by analysing the road information. Then, the extracted image information is comprehensively analysed, selected and reorganised and combined with the road features. Finally, it is fused with the structural relationship, model and road-related rules of the road elements to identify the road.

### 3.1.1. Main Methods

Numerous road extraction algorithms based on high-resolution images have been developed over the past few decades, making it difficult to classify them. Traditional methods include automatic and semiautomatic extraction. Some methods based on deep learning have emerged and attracted considerable attention, owing to their high precision, in recent years. In this paper, we summarised the heuristic and data-driven road extraction methods based on two state-of-the-art reviews [34,71]. A comparison between the different data-driven methods applied on the Massachusetts road dataset is shown in Table 4. It can be seen from Table 4 that most data-driven methods can obtain a high precision (better than 0.8). Meanwhile, different methods have advantages and disadvantages.

**Table 4.** Comparison of different data-driven methods on Massachusetts road dataset.

| Method | Advantages | Disadvantages | References | Precision |
|---|---|---|---|---|
| Patch-based DCNN | Weight sharing, less parameter | Inefficiency, large-scale training samples | [168], 2016 [38], 2017 | 0.905 0.917 |
| FCN-based | Arbitrary image size, end to end training | Low fitness, low position accuracy, lack of spatial consistency | [36], 2016 | 0.710 |
| DeconvNet-based | Arbitrary image size, end to end training, better fitness | High cost of computing and storage | [49], 2017 [51], 2018 | 0.858 0.919 |
| GAN-based | More consistent | Non-convergence, gradient vanishing, and model collapse | [65], 2017 [66], 2017 | 0.841 0.883 |
| Graph-based | High connectivity | Complex graph reconstruction and optimisation | [169], 2018 [170], 2020 | 0.835 0.823 |

The heuristic extraction methods can be subdivided into automatic and semiautomatic methods according to the degree of automation facilitated by the method. The semiautomatic extraction algorithms require initial seeds, and the user needs to check the results frequently. In these methods, the seeds and directions should be provided manually, and the algorithms can recognise and roll back the previous results. Several classic semiautomatic methods exist, such as the active contour model, dynamic programming, geodesic path and template matching. The active contour model, proposed by Kass et al. [171] and also known as the balloon snakes or ribbon snakes model, can extract road information through contour deformations from the labelled lines or points. The geodesic path method can produce a road-probability map through the extracted road edges. The dynamic programming method requires road parameters and mainly focuses on solving optimisation problems [172]. The template matching method can extract the road features through constructed windows and then matching the extracted points. All these semiautomatic methods include automatic processes.

Automatic road extraction uses information such as road topology and context features to extract and recognise roads through methods such as pattern recognition, computer vision, artificial intelligence, and image understanding [173–177]. Even though there are no fully automatic algorithms for all types of high-resolution images, several methods [71] such as segmentation-based, edge analysis, map-based, swarm intelligence-based, object-based and multispectral segmentation methods are much more efficient than the semiautomatic methods. Methods based on segmentation can identify regions through numerous algorithms [71] such as support vector machines (SVMs), artificial neural networks, Bayesian classifiers, mean shifts, watershed algorithms, super pixel segmentation, Gaussian mixture models, graph-based segmentation and conditional random field (CRF) models, which are usually used in combination to improve the classification accuracy. The edge analysis method is realised via edge detection, which is suitable for extracting the main roads. The map-based methods, especially OpenStreetMap, focus on urban roads. Swarm intelligence-based methods such as ant colony optimisation, artificial bee colony and firefly algorithms use discretisation and networking to simulate real biological behaviours [178]. Object-based methods are used to classify objects through object segmentation and feature characterisation. Multispectral segmentation methods usually require the support of multispectral or hyperspectral images [2].

With the application of deep learning techniques in road extraction, many data-driven methods such as patch-based CNN models, FCNs, deconvolution networks, GAN models and graph-based methods have been proposed for road extraction [34,71]. The patch-based CNN method can exploit a large image and predict a small patch of labels through the CNN models based on structured and refined CNNs; however, this method is time-consuming and computationally inefficient [34]. The FCN method predicts images by replacing connected layers with output labels and classifies images at the pixel level using the FCN-32 and U-shaped FCN models, thereby improving the road extraction accuracy [44,45]. A variant of FCN—DenseNets—including SegNet, DeepLab, RCFs, Y-net and U-Net decoder [49,51,52,54] can extract hierarchical features from images, especially

high-resolution images. The GAN method includes a generator and discriminator to segment images and distinguishes between forged and real images, respectively [179]. The graph-based method realises the vector representation of road maps through iterative road tracking and polygon detection strategies.

### 3.1.2. Status and Prospects

Each algorithm has its advantages and disadvantages. For example, Lian et al. [71] compared the heuristic and data-driven road extraction methods based on four public datasets. Their results indicated that data-driven methods could achieve more than 10% accuracy compared to that achieved using heuristic methods. Similar conclusions were reported in [34]. However, data-driven methods have limitations such as the requirement of large training samples, long processing time and high-speed computing (most algorithms require graphics processing units). In addition, the training parameters used in one dataset may not be able to achieve high accuracy when used in another dataset. In contrast, some heuristic methods require less time and can meet real-time demands in some applications.

Road extraction using the currently available high-spatial-resolution images has some challenges. The key step in road extraction from high-resolution images is to describe the road features. Describing the linear or narrow bright band of a road, which provides good detection results, is the main focus of most existing methods. However, with the improvement in image resolution, we can obtain more noise interference (shadows, buildings and road obstructions) and more detailed road features; therefore, the road objects can also be described more precisely in this case [34,173]. Furthermore, road objects include many complex phenomena such as occlusion or shadows, discontinuities, sharp bends and near-parallel boundaries with constant widths. Incorporating all these factors and modelling them into a single model is almost impossible. Therefore, it is essential to establish a multimode approach to extract roads from images with high spatial resolutions.

### 3.2. Road Extraction Based on Hyperspectral Images

Multispectral remote sensing images have been used in road extraction because of their high spatial resolution and multiple spectral features. The commonly used data sources are the satellite multispectral images including QuickBird, IKONOS [180,181], Worldview 2 satellite [182], Landsat satellites [183] and Gaofen 1 and Gaofen 2 satellites. In addition, because of the large number of bands (generally more than 100 bands) and continuous spectrum bands, hyperspectral images are used for conventional road extraction as well as show great potential for road condition detection, road material identification, road pothole detection and crack detection.

### 3.2.1. Main Methods

Most road extraction methods using multispectral images are based on high-spatial-resolution images, and these methods can also be divided into heuristic and data-driven methods. Similar to the road extraction from high-resolution images, the heuristic methods in this case can also be divided into semiautomatic and automatic categories; some applications of these heuristic methods are reported in [184–189]. However, few data-driven methods are exclusively used for road extraction based on multispectral images, owing to the lack of public datasets. There are fewer road extraction methods based on hyperspectral images than on high-resolution images, and some of them are included in hyperspectral classifications. In this paper, we introduce road extraction methods using hyperspectral images based on different platforms.

The spaceborne hyperspectral imagers can realise a larger swath and provide a better stability platform than do the airborne instruments; hence, the spaceborne hyperspectral imagers are suitable for large-area work. However, they are mainly used to identify and extract arterial roads such as highways because of their low spatial resolution. The Hyperion hyperspectral imager of the US—EO-1 satellite—is the currently available spaceborne hyperspectral sensor for road extraction. Sun [190] used Hyperion hyperspectral data to

complete the road network extraction from an image using three steps: road searching, road tracking and road connecting. In road searching, the spectral information in the image is used to find road features, and several different road feature extraction methods are qualitatively compared; however, no quantitative extraction accuracy results are provided in Sun's report.

Airborne hyperspectral imagers can achieve higher spatial and spectral resolutions than do the spaceborne platforms; however, their operating efficiency is lower than that of the spaceborne instruments. They are mainly used for the extraction of urban roads and detection of road conditions. Airborne hyperspectral instruments currently used for road extraction mainly include AVIRIS, CASI, HyMap, HYDICE and AsiaFenix [122]. In 2001, Gardner et al. [191] used the hyperspectral dataset of AVIRIS in Santa Barbara, USA, to map the different types of typical urban surface features through multiterminal spectral analysis. Then, the Q-tree filter was used to distinguish between the roofs and the roads constructed using the same materials and exhibiting similar spectra. The visible results showed that the main roads in the image can be preliminarily extracted; however, several shortcomings were observed in the extraction of roads blocked by vegetation and connectivity of the road network. Noronha et al. [192] used the urban hyperspectral dataset of AVIRIS and a spectral database, based on the surface materials of the main urban features collected in the field, to extract the road centreline and observe the road surface conditions. Furthermore, the optimal parameters for designing a multispectral instrument to extract urban land-use types was proposed based on these reported results. The overall classification accuracy and kappa coefficient were 73.5% and 72.5%, respectively [192]. Based on the airborne hyperspectral image data of HYDICE and HyMap, Huang and Zhang [193] used an adaptive mean-shift method to accurately classify six major urban features in the image, including roads, houses, and grass. The overall classification accuracy was above 97%, and the road classification accuracy was above 95%. Resende et al. [194] used CASI-1500 airborne hyperspectral data to study the extraction of asphalt roads in cities. The results based on ISODATA unsupervised classification and maximum likelihood supervised classification methods qualitatively showed that the hyperspectral image could be used to extract the main asphalt roads in the city; however, they did not report the extraction accuracy. In 2012, Mohammadi [195] used HyMap airborne hyperspectral image data to study the classification of materials used in urban roads and the state of asphalt road conditions. He mainly distinguished asphalt roads, cement roads, and gravel roads, and based on this result, reported three road conditions: good, medium and poor asphalt roads. However, the experimental results were limited by the spatial resolution of the dataset, and a large number of unclassified pixels were not analysed by the method. Therefore, further studies are required to improve the methods used for reducing the number of unclassified pixels.

Currently, some publicly available airborne hyperspectral datasets such as Pavia Centre and University area hyperspectral datasets and Indian Pine hyperspectral datasets [196,197] are also used for road classification and recognition. In 2012, Liao et al. [196] proposed a directional morphology and semi supervised feature extraction to classify three hyperspectral datasets. The classification accuracy of the roads was the highest, reaching more than 97%; however, this classification accuracy was closely related to the number of training samples and extracted features. Miao et al. [197] studied the extraction of road centrelines from high-resolution images based on shape features and multiple adaptive regression splines. This method combined the shape features and spectral information to extract road segments from high-resolution images and then used multivariate adaptive regression spline functions to extract the road centrelines. The method was applied to the Pavia Centre hyperspectral dataset, and an extraction accuracy of 99% was obtained for the road centreline extraction. This method was based on uniform surface properties; hence, it was suitable only for high-resolution images and not for low-resolution images. In addition, the main limitation of this method was that the threshold in the method must be determined manually.

In addition to spaceborne and airborne hyperspectral imagers, UAV hyperspectral imaging systems that have gradually emerged in recent years have garnered increasing attention owing to their low cost and high spatial resolutions. These systems are mainly used for road condition detection and road material identification in specific areas [15,198]. However, their operating efficiency is lower than that of the spaceborne and airborne platforms, because of their low flight altitude. A summary of road extraction using hyperspectral images is shown in Table 5, which indicates that hyperspectral imaging systems of different platforms have different characteristics for road extraction due to spatial resolution. Spaceborne hyperspectral imagers are primarily used to extract main roads, while airborne hyperspectral imagers can be used for road quality assessment and road condition monitoring.

**Table 5.** Summary of road extraction using hyperspectral images.

| Method | Platform | Characteristic | References |
|---|---|---|---|
| Traditional process includes the spectral information | Spaceborne | Extract the main roads | [190], 2003 |
| Spectral mixture and Q-tree filter | Airborne | Assess road quality | [191], 2001 |
| Pixel to pixel classification | Airborne | Extract asphalted urban roads | [194], 2008 |
| Spectral angle mapper | Airborne | Road classification and condition determination | [196], 2012 |
| Computing the angle from spectral response | UAV | Detect pavement roads | [198], 2019 |

## 3.2.2. Status and Prospects

Only a few reports on extracting road information from spaceborne hyperspectral images are available. This is because the spatial resolution is insufficient to extract road information accurately (e.g., the spatial resolution of Hyperion and Gaofen-5 hyperspectral imagers is 30 m), especially for urban roads and narrow roads. In addition, spaceborne hyperspectral images are difficult to acquire, and public datasets for road extraction are not available. Airborne hyperspectral images are still the main data source for the study of road extraction, because of their higher spatial resolutions and lower costs compared to those of the spaceborne data.

Hyperspectral images have shown considerable potential for road condition detection, road material identification, road pothole detection and crack detection. However, the preprocessing accuracy of the hyperspectral images should be improved to promote their applications. Hyperspectral data with geometric correction and relative radiometric calibration can meet this requirement for qualitative applications such as road detection. However, for quantitative applications such as pavement material recognition, more steps are required, such as high-precision absolute radiometric calibration, spectral calibration and atmospheric correction.

## 3.3. Road Extraction Based on SAR Images

### 3.3.1. Main Methods

In general, roads in SAR images appear as dark linear features. However, the differential orientation of roads and antennas influences the ability of SAR to identify roads. In traditional heuristic methods, road segments are often extracted using an edge/line detector. Then, a graph is generated from the segments and optimised, and the segments are connected to develop a coherent road network. Recently, data-driven deep learning-based semantic segmentation methods have been reported. Moreover, road junctions and bridges are important parts of road networks and have been the topic of some previous studies.

Road network extraction from SAR images has been reported in various studies [199–201]. In [200], two local line detectors were used, and the results were fused to find candidates for road segments. The road segments were connected using a Markov ran-

dom field, and an active contour model (snake) was used for the postprocessing. In [200], this method was applied on dense urban areas, and very-high-resolution data and different flight directions were combined to improve the results. In [201], constant false alarm rate detection, morphological filtering, segmentation and Hough transformation were integrated to recognise roads in high-resolution polarimetric airborne SAR images. The strong backscattering from fences was used to detect bridges, and then the roads were recognised using a Hough transformation

The fusion of different SAR datasets and algorithms in early studies improved road extraction accuracy. In [202], different preprocessing algorithms, road extractors and different images of the same area were fused, and a multiscale approach was used for road extraction. For SAR, a combination of different view angles is also proposed. Lisini [203] extracted road networks by combining the line extractor results with two classification results. Hedman [204] detected rural and urban areas and then fused a road extractor for rural areas to develop an extractor designed for urban areas.

The latest reported studies used new high-resolution data, usually from the TerraSAR-X or GaoFen-3 satellites. In [205], multiscale geometric analysis was performed on vectorised detector responses for road network grouping using two TerraSAR-X datasets. In [206], a method suitable for analysing SAR images of different resolutions was proposed. A weighted ratio line detector was developed to extract the road ratio and direction information. The road network was constructed using a region-growing method and tested using four SAR datasets obtained from different study areas. Saati [207] extracted road networks based on a network snake model and three TerraSAR-X images. Xu [208] introduced an algorithm in which road segment extraction and network optimisation were performed simultaneously using a Bayesian framework, multiscale feature extractor and CRF; for this analysis, Xu used the TerraSAR-X and airborne SAR data. Xiong [209] proposed a method based on vector Radon transformation, and promising results were presented for six SAR images of different resolutions, bands and polarisations; these images were obtained from airborne SAR, GaoFen-3 and TerraSAR-X. In general, for road extraction from new very-high-spatial-resolution (VHR) data, completeness of 74–93% and correctness of 70–94% have been reported, depending mostly on the study site.

Interferometric information has been used in road extraction by Jiang et al. [210], and the best results were obtained by the fusion of intensity and coherence information. Roads were considered as distributed scatterers and were separated from permanent and temporally variable scatterers. A constant false-alarm-rate line detector based on Wilks' test statistics has been proposed by Jin et al. [211] for polarimetric SAR images.

Deep learning-based methods have been the topic of a few recent studies; they can be used to extract initial road information with good quality for network construction. Zhang [158] compared U-Net (FCN) and CNN to machine learning methods using dual-polarisation Sentinel-1 data; he reported that VV polarisation was better than VH, and dual-polarisation data was better than the single-polarisation data for road extraction. F1 score of 94% was achieved (the same area for training and testing but different shuffled samples were used) using the U-Net and dual-polarisation data. This F1 score was better than that of the best machine learning method (random forest) by 5%. Henry [46] enhanced the fully CNN (FCNN) sensitivity for thin objects and compared the FCN-8s, U-Net and Deeplabv3+ methods for road segmentation. The sensitivity was increased by addressing class imbalance in training and using spatial tolerance rules. A summary of road extraction using SAR images is shown in Table 6. It can be seen that the precision in diverse situations is quite different. Appropriate methods or a combination of multiple methods should be considered according to the scene's characteristics.

**Table 6.** Summary of road extraction using SAR images.

| Method | Category | Characteristic | References | Precision |
|---|---|---|---|---|
| Multiple Detectors | Heuristic | Fusion of different pre-processing algorithms, road extractors | [202], 2003 | 0.580 correctness |
| Line based on vector Radon transform | Heuristic | Suitable for different platform SAR images | [209], 2019 | 0.700–0.940 correctness |
| Multitemporal InSAR covariance and information fusion | Heuristic | Use interferometric information | [210], 2017 | 0.816 correctness |
| FCN-based | Data-driven | Automatic road extraction | [158], 2019 | 0.921 |
| FCN-8s | Data-driven | Lack efficiency | [46], 2018 | 0.717 |

### 3.3.2. Status and Prospects

Automatic road extraction from SAR imagery remains solely experimental to date. For operational tasks, semiautomatic methods [172] with human intervention or manual postprocessing are required. Large-scale tests have not been conducted, and for the reported methods, remarkable variations in the results between different test sites have been observed. The studied road types include forest roads, city streets, highways, desert roads, gravel roads, paved roads, icy roads and bridges. However, different or mixed road classes have been evaluated in only a few studies. Different methods are required for different road types and study areas. In general, the developed methods are complex and involve many steps.

Owing to the side-looking sensor [212], SAR produces many linear features, causing many false alarms in road extraction. In addition, road detection varies with the looking direction, especially in urban areas (radar shadows). The speckle and speckle filters also affect the road extraction accuracy. In addition, the roads may appear bright because of the surrounding structures, instead of appearing as dark lines. Therefore, the road appearance differs depending on the SAR image resolution. A highly complex image is obtained with high-resolution data, showing strong geometric effects in urban areas. Therefore, the use of multiple datasets is often required. Conversely, road detection from SAR is independent of road surface material; this feature is less pronounced in the optical detection method. Thus, the fusion of SAR and optical signals can be beneficial.

### 3.4. Road Extraction Based on LiDAR Data

#### 3.4.1. Main Methods

Ground points and nonground points can be easily distinguished based on pulse and elevation information in airborne LiDAR data. To classify ground and road/nonroad areas, intensity information is needed. Roads, such as water surfaces, have low intensity. In addition to the surface reflectance, the intensity values are affected by atmospheric attenuation, transmitted power, detection range and incidence angle. Therefore, calibration is required to apply the methods over large areas. Road detection using intensity is usually based on the surface homogeneity and consistency of roads. The LiDAR-based methods for road extraction either use point cloud processing and classification or are based on a digital surface model (DSM), digital terrain model (DTM) and intensity raster produced from the point clouds. In some reported studies, the main focus was on data classification, while in others, a complete road network was extracted.

Point cloud-based automatic road extraction using LiDAR height and intensity was first proposed by Clode [213]; in this study, a hierarchical classification method was used to classify point clouds progressively into roads and nonroads. Individual points were selected based on the height difference to generate a DTM, and the intensity value was obtained via filtering based on point density and morphological filtering of a binary image. The method was enhanced in [214], where a phase-coded disk algorithm was introduced to vectorise the binary road network image. Hu [215] proposed a method for road centreline extraction using the salient linear features observed in the images of complex urban areas; in this method, tensor voting was used to eliminate the nonroad areas.

Intensity variations in the road network were considered in [216], where road points were selected from ground points based on a local intensity distribution histogram and

filtered by roughness and area. Hui [217] extracted road centrelines using three steps; a skewness-balancing algorithm was proposed to obtain the intensity threshold. A rotating neighbourhood algorithm was proposed to extract the main roads (by removing the narrow roads), and a hierarchical fusion and optimisation algorithm was proposed to extract the road network. For the three test sites, correctness values of 43–92% and completeness of 36–91% were obtained.

A raster-based road extraction method for grid-structured urban areas was proposed by Zhao [218]; in this study, the ground objects were classified, road centrelines were extracted using a total least square line fitting approach, and a voting-based road direction was used to evaluate each road segment's reliability by removing areas such as parking lots from the road segments. For a complete road network extraction, different parts of the road network such as junctions and bridges need to be considered as well. Chen [219] proposed an automatic method to detect and delineate road junctions from rasterised ALS intensity and normalised DSM in three steps: roughness-enhanced Gabor filters for key point extraction, a higher-order tensor voting algorithm to find the junction candidates, and a geometric template matching to identify the road junction positions and road branch directions. A bridge detection algorithm was proposed by Sithole [220]; in this method, DTM cross-sections, that is, profiles, were used to identify bridges. Moreover, forest road detection is possible using detailed ALS DSMs.

Road details and complex structures can be extracted from very dense point clouds as well and modelled. In [221], road points were accurately labelled from a dense point cloud of an urban area, using an approximate 2D road network map as the input. They combined both large-scale (snake smoothness) and small-scale (curb detector) cues to extract roads. The method worked on all types of roads, including tunnels, bridges and multilevel intersections. In [222], UAV data was used to perform fine-scale road mapping. Soilan [223] studied the automatic extraction of road features (sidewalks, pavement areas and road markings) from high-density ALS point clouds.

3D modelling is required for the most complex parts of the road network such as overpasses and multilevel intersections. The use of LiDAR point clouds to derive 3D city models was reviewed in [77]. Cheng [160] studied the detailed 3D reconstruction of multilayer interchange bridges, and satisfactory results were obtained for very complex bridges.

In several studies, road detection has been carried out as a part of land cover mapping. Urban land cover mapping by integrating rasterised LiDAR height and intensity data at the object level was proposed by Zhou [224]; for this method, an accuracy similar to that of multispectral optical imagery accompanied by ALS DSM was obtained. Matkan [225] classified the point cloud into five land cover classes using SVM; during the postprocessing, gaps in the road network were located and filled using a method based on Radon transformation and spline interpolation.

Land cover classification using multispectral ALS (MS-ALS) has been the topic of several recently reported studies. Even though a complete road network was not extracted, the road classification results were promising [164,226]. In [226], the point-based classification completeness for roads was 86% and 92%. In the raster-based classification, accuracies of 92% and 86% were obtained. Karila [227] used rasterised MS-ALS data for a typical road surface (that is, asphalt and gravel) and classified the road types from highways to cycle ways. Due to the lack of shadows, more complete roads (80.5%) were retrieved using this method than using the optical aerial images (71.6%). Ekhtari [228] classified multispectral point clouds into 10 land cover classes using an SVM. Three types of asphalt and concrete classes were included. In general, slightly better results were obtained when the classification was carried out in the point cloud domain; however, the computational costs increased significantly.

Deep learning methods for MS-ALS have been explored in a few recent studies. Pan [229] used deep learning-based high-level feature presentation (deep Boltzmann machine) and machine learning methods for land cover classification. Pan [230] proposed

a CNN-based classification approach for MS-ALS data. The classification accuracy and computational performance of the constructed CNN model were superior to those of the classical CNN models. Yu [231] proposed a hybrid capsule network using MS-ALS data. The data were rasterised based on the elevation, the number of returns and intensity of the three channels, and an accuracy of 94% was obtained for the road classification. Dense point clouds from SPL were used for land cover classification in [232]. Due to the rough appearance of the intensity images created from the SPL data, small features such as narrow roads were often difficult to distinguish in the intensity images. A summary of road extraction using LiDAR data is shown in Table 7. Notably, ML-ALC has become the main approach for road extraction in recent years.

**Table 7.** Summary of road extraction using LiDAR data.

| Method | Category | Characteristic | References | Correctness |
|---|---|---|---|---|
| Hierarchical fusion and optimisation | ALS | Extract road centreline | [217], 2015 | 0.914 |
| Point-based classification Raster-based classification | MS-ALS | Land cover classification | [226], 2017 | 0.920 0.860 |
| Object-based image analysis and random forest | MS-ALS | road detection and road surface classification | [227], 2017 | 0.805 |
| Support vector machine | MS-ALS | Three types of asphalt and a concrete class | [228], 2018 | 0.947 (Overall accuracy) |
| Hybrid capsule network | MS-ALS | Land cover classification | [231], 2020 | 0.979 (Overall accuracy) |

### 3.4.2. Status and Prospects

Airborne LiDAR data are used in the 3D modelling of city road networks [77]. ALS provides direct 3D information for road extraction and is less affected by occlusions and shadows than do optical data. In addition, ALS provides 3D road information with elevation; this is especially useful in complex interchange areas. However, the area covered during the flight considerably limits the automatic road extraction process using ALS. Extensive investigations are still required to address the underlying issues, i.e., the development of fully automatic algorithms suitable for various landscape and road types, application of intensity data over large areas, reducing the number of false positives (car parks, squares, playgrounds, etc.) and identifying as well as connecting road segments shadowed by occlusions (e.g., trees and vehicles). Road markings must be taken into consideration in VHR. The national ALS datasets provide a good basis for mapping; however, these datasets are seldom updated. LiDAR sensors mounted on mobile mapping systems, UAVs or VHR satellite imaging instrument can be used for map updating. In addition, the new MS-ALS data [227,228] and new dense point clouds created by collecting single photons enable the automatic detection of roads with higher accuracy.

## 4. Combination of Multisource Data for Road Extraction

### 4.1. Combination of High-Resolution Images with Other Data for Road Extraction

High-resolution images reveal very fine details of the earth's surface and geometry; however, high-resolution imaging also increases the geometric noise and only provides spectral and spatial information on the surface. LiDAR data, as a special data source, can provide 3D information about objects. Tiwari et al. [233] proposed automatic road extraction methods through an integrated approach involving ALS altimetry and high-resolution imaging. The method was used to extract road information without background objects, and showed an increased road extraction accuracy of 90% when applied to Amsterdam data. Hu et al. [234] proposed a grid-structured urban road network extraction method using LiDAR data and high-resolution imagery. A significant improvement in the road extraction accuracy was obtained using this method than using high-resolution imagery or LiDAR data. Zhang et al. [235] proposed a method to improve the accuracy of road centreline extraction using high-resolution images and LiDAR data. The method adopted the minimum area bounding rectangle-based interference-filling approach, multistep approach

and Harris corner detection. The experimental results based on the datasets of Vaihingen, New York, and Guangzhou showed that the proposed method was efficient in identifying complex scenes.

### 4.2. Combination of Hyperspectral Images with Other Data for Road Extraction

Feng et al. [236] fused hyperspectral images and LiDAR data to map urban land use using a state-of-the-art CNN. To improve the speed of the network design, the same structure was used in both the hyperspectral and LiDAR branches. Each branch used a residual block to extract multiscale, parallel, and hierarchical features. The experimental results underlined the efficient road extraction performance of the proposed method. When only hyperspectral images and LiDAR data were used, the highway classification accuracies were found to be 65.35% and 42.08%, respectively. However, this highway classification accuracy increased to 80.89% when fused data was used. Elaksher et al. [237] combined the LiDAR-based hyperspectral images obtained from the AVIRIS and DEM. First, a vector layer of polygons was constructed using the DEM data. Second, the buildings in the hyperspectral images were removed, and then the road and water were classified using a supervised classifier. The experimental results demonstrated that the performance of this classification process could be improved by using LiDAR data to remove the buildings from the hyperspectral image before the classification. The detection rate of the road was 91.3%, and the false alarm rate was 0. Two examples of multiple remote sensing data fusion are presented in [238]. One is the fusion of hyperspectral images with SAR images; this fused data can improve the detection accuracy of the target. The other is the fusion of hyperspectral images with high-resolution images. The spatial–spectral information of the target was fully analysed using the two combined data sources, and the identification accuracy was improved considerably.

### 4.3. Combination of SAR Images with Other Data for Road Extraction

Several studies have been published on the fusion of optical imagery and SAR data. Cao [239] proposed road extraction via the fusion of infrared and SAR images. Lin et al. [240] compared multiple remote sensing datasets (Spot5, IKONOS, QuickBird, DMC and airborne SAR datasets) and algorithms. Road trackers were designed for five different road types: national highways, interstate highways, railroads, avenues and lanes. Evidently, fused multiple remote sensing data were more efficient than a single data source. Perciano et al. [241] fused TerraSAR-X and QuickBird data for road network extraction at two test sites, and the road extraction accuracy for the fused data was 10–30% higher than those obtained for individual datasets. Multitemporal SAR image stacks (TSX and CSK) were also studied. Bartsch et al. [242] studied the arctic settlements using Sentinel-2 optical and Sentinel-1 SAR satellite images. Pixel-based classification using a gradient boosting machine and a deep learning approach based on windowed semantic segmentation using U-Net architecture were compared. Asphalt roads were easily detected than gravel roads, and for arctic mapping, both methods and sensors were recommended. Liu et al. [243] studied urban area mapping using Sentinel-1 SAR and Sentinel-2 optical data and proposed the integration of object-based postclassification refinement and CNNs for land cover mapping. Notably, for road mapping, SAR backscattering provided different physical information on roads (low backscatters) than that provided by optical remote sensing; moreover, the roads were identified with higher accuracy by combining the optical data with that of the SAR. Lin et al. [244] extracted impervious surfaces using optical, SAR and LiDAR DSM data. The non-shadow and shadow classes were trained using the combined optical–SAR–LiDAR data. As a result, the shadow effects in the classification results were reduced.

### 4.4. Combination of LiDAR with Other Data for Road Extraction

Aerial imaging cameras are often accompanied by LiDAR sensors in airborne mapping systems. Thus, it is common to fuse LiDAR point clouds and optical aerial imagery. In addition, optical satellite data are used as well. In particular, additional colour information

is useful for single-channel LiDAR. Segmentation of the input imagery is often performed to enable object-based fusion of the datasets.

Kim [245] proposed a method to improve the classification of urban areas by fusing high-resolution satellite images (WorldView-2) and ALS data. Special attention was paid to the elevated roads, which were first detected in LiDAR ground points. Then, buildings were detected, and supervised SVM classification was performed on areas without elevated roads or buildings. Liu [246] proposed a road extraction framework based on the fusion of ALS point clouds and aerial imagery; in this framework, pseudo scan lines were created from the fused data, and a rule-based edge-clustering algorithm was used to extract the road segments. Mendes classified road regions using an ANN by integrating aerial RGB (where R is red, G is green and B is blue) images, laser intensity and height images. Compared to the use of optical data alone, incorporating the laser intensity data helped to overcome the road obstructions caused by shadows and trees, and the height information helped in separating the aboveground objects from the ground objects. Zhang [235] proposed an object-based method for road centreline extraction from aerial images and ALS DSMs; using this method, a completeness and correctness of over 90% was obtained for two of the test data sets, and a completeness and correctness of over 80% was obtained for a third large site. Further developments were proposed for curved roads.

### 4.5. Some Scopes of Future Research in Road Extraction

A summary of road extraction based on different data sources is presented in Table 8. Some prospects can be derived based on the status of current road extraction from high-resolution images. First, data-driven road extraction methods exhibit excellent performance and high extraction accuracy [34]. Therefore, more robust data-driven methods should be developed and verified using different datasets. In addition, it is difficult to obtain high detection accuracy using only one algorithm in some cases; therefore, a combination of multiple road extraction methods must be studied [247]. Finally, the combination of data sources (e.g., hyperspectral, SAR and LiDAR) should be evaluated further. Spatial resolution is one of the most important factors affecting the performance of road extraction methods. High-spatial-resolution images can describe fine objects in detail. However, this increases the spectral variability within the class [248]. Wang et al. [249] demonstrated that images with a spatial resolution of 0.5 m had higher accuracy than those with a spatial resolution of 0.1 m. Data-driven methods show higher road extraction accuracy with improved spatial resolution than do the traditional heuristic methods [34,71].

**Table 8.** Summary of road extraction based on different data sources.

| Data | Resolution/ Mapping Unit | Extent | Advantages | Roads Extracted Mostly by |
|---|---|---|---|---|
| High spatial resolution [71], 2020 | 0.5–10 m | Local/regional/global | Most tools available, "basic" software | Colour, texture |
| Hyperspectral [198], 2019 | 0.25–30 m/ (>100 channels) | Local/regional | Spectral information | Colour, texture and spectral features |
| SAR [72], 2014 | 1–10 m | Local/regional/global | See through clouds, rapid mapping | Linear features/edge |
| ALS [75], 2017 | 0.25–2 m | Local (nationwide) | Height information | 3D geometry (intensity) |

It is necessary to continue to study road extraction from airborne hyperspectral images [80]. In addition, new road extraction methods based on hyperspectral images should be developed. In particular, the advantages of a large number of bands and continuous spectrum coverage of hyperspectral images should be further exploited, and new data-driven methods should be proposed [250]. Finally, hyperspectral image datasets with a wide coverage area and road labels should be produced to promote the road extraction application of hyperspectral data.

Recently, the open-access global SAR satellite datasets (Sentinel-1) have enabled the mapping and monitoring of large areas. However, the resolution of this method is limited. Microsatellite constellations can acquire large amounts of very-high-resolution data with higher frequency; however, this aspect has not been studied in detail. For ALS, deep learning methods [251] for image classification are rapidly emerging, followed by methods for earth observation. Publicly available open training datasets acquired by the earth observation satellite are being used for road extraction. The highest accuracies for road classification have been reported using deep neural network algorithms. However, these studies cover limited areas where they perform relatively well, but no large-scale tests have been conducted yet.

## 5. Conclusions

High-resolution and hyperspectral images have been widely used in digital road network extraction. More satellites with high spatial resolution and short revisit periods are being developed and launched, promoting the development of heuristic and data-driven road extraction methods. However, few of these hyperspectral satellite data can be used for road extraction. Therefore, airborne systems are still the main approach for the acquisition of hyperspectral data [80,252]. Data-driven methods have high accuracy and show significant potential; however, transfer learning needs to be improved. In addition, the combination of high-resolution image data with other data sources such as LiDAR is one feasible approach to solve some challenging issues such as occlusion or shadows.

Large areas can be rapidly mapped using weather-independent spaceborne SAR images; for example, images acquired after sudden changes in the target area. In addition, global datasets enable global mapping. However, the roads are challenging to interpret because of the interaction of the signal with the surrounding areas. ALS provides excellent data for the generation of topographic databases and detailed mapping of limited areas. Multispectral ALS may be the best remote sensing data source for road mapping; however, only small areas are covered at a time in this case. Further studies are required for developing sensors for various landscapes and road types; moreover, fully automated road detection is still in its infancy.

High-resolution imaging, hyperspectral imaging, SAR imaging, and LiDAR are currently the primary techniques for road extraction. As shown in Table 8, different data sources have unique characteristics. For example, high-resolution images have high spatial resolution and contain rich textures, shapes, structures, and neighbourhood relations of ground objects. Hyperspectral images have multiple data dimensions and rich spectral features. In addition, SAR imaging and LiDAR are less affected by external environmental factors such as clouds, fog, and light and can operate in all weather conditions. Combining different remote sensing data to use their respective advantages is a notable approach for developing advanced road extraction methods in the future.

**Author Contributions:** Bibliographic review, J.J., H.S., C.J., K.K. and M.K.; paper organisation, Y.C. and J.J.; drawing of conclusions, J.J.; writing—original draft preparation, J.J., H.S., C.J., K.K. and M.K.; writing—review and editing, E.A., E.K., P.H., C.C. and T.X.; supervision, Y.C.; project administration, J.H.; funding acquisition, Y.C. and T.W. All authors have read and agreed to the published version of the manuscript.

**Funding:** This research was financially supported by Academy of Finland projects "Ultrafast Data Production with Broadband Photodetectors for Active Hyperspectral Space Imaging (No. 336145)", Forest-Human-Machine Interplay-Building Resilience, Redefining Value Networks and Enabling Meaningful Experiences (UNITE), (No. 337656) and Strategic Research Council project "Competence-Based Growth Through Integrated Disruptive Technologies of 3D Digitalization, Robotics, Geospatial Information and Image Processing/Computing–Point Cloud Ecosystem (No. 314312). Additionally, the Chinese Academy of Science (No. 181811KYSB20160040 XDA22030202), Shanghai Science and Technology Foundations (No. 18590712600) and Jihua lab (No. X190211TE190) and Huawei (No. 9424877) are acknowledged.

**Institutional Review Board Statement:** Not applicable.

**Informed Consent Statement:** Not applicable.

**Data Availability Statement:** Not applicable.

**Acknowledgments:** The authors would like to thank the contributions of the editor and reviewers.

**Conflicts of Interest:** The authors declare no conflict of interest.

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
