# Peer review of "Review on Active and Passive Remote Sensing Techniques for Road Extraction"

_remotesensing, doi:10.3390/rs13214235_

Round 1
Reviewer 1 Report
GENERAL OBSERVATIONS 1. Minor English revision is required across the paper. For example: - line 167 - the UVA abbreviation should read UAV, - line 182 - the sentence is similar to line 166 and could be removed. 2. The paper is missing some important remote sensing techniques for road extractions such as Ground Moving Target Indicator (GMTI), GPS data and smart phones, or Street View. Recommend adding some of these techniques to the paper. SPECIFIC OBSERVATIONS 3. Line 33 - Suggest to include a discussion in the introduction section about the "so what" of remote sensing road extraction in comparison with traditional cartographic techniques. 4. Line 208 - When discussing the SAR virtual long antenna and the spatial resolution it is worth mentioning that the long antenna aperture is realised in the cross-range direction and that the range resolution is given by the pulse width (in the general the bandwidth of the signal). Another particularity of the SAR compared with optical sensors is the low correlation between range and spatial resolution. 5. Line 225 - Change detection is an important technique used in the SAR domain and numerous papers are available which could be referred to here.
Author Response
Dear reviewer, thank you for your comments concerning our manuscript. Those comments are all valuable and very helpful for revising and improving our paper and the important guiding significance to our research. We have studied all comments carefully and tried our best to improve the manuscript, and made some changes to the manuscript. These changes will not influence the content and framework of the paper.
Comment
GENERAL OBSERVATIONS
- Minor English revision is required across the paper. For example: - line 167 - the UVA abbreviation should read UAV, - line 182 - the sentence is similar to line 166 and could be removed.
Reply
Thank you for your kind reminder. We have fixed them. In addition, we rechecked the manuscript and revised some minor errors.
Comment
- The paper is missing some important remote sensing techniques for road extractions such as Ground Moving Target Indicator (GMTI), GPS data and smart phones, or Street View. Recommend adding some of these techniques to the paper.
Reply
Thank you for your suggestions. We have added these techniques and relevant citations to the Introduction section:
Remote sensing data used for road extraction include ground moving target indicator (GMTI) tracking, smart phones global positioning system (GPS) data, street view images, synthetic aperture radar (SAR) images, light detection and ranging (LiDAR) data, high-resolution images, and hyperspectral images. GMIT radar has been used for extracting road map information due to the advantages of all-weather, real-time capabilities, and wide-area [8,9]. Smart phones GPS data were used for extracting road centrelines and monitoring road and traffic conditions [10,11]. Street view images obtained by Google and Baidu companies have been used for the detection, classification, and mapping of traffic signs and road crack information extraction [12,13].
This paper mainly studied road extraction based on high-resolution images, hyperspectral images, SAR images, and LiDAR data. Therefore, other remote sensing techniques for road extractions such as Ground Moving Target Indicator (GMTI), GPS data and smart phones, or Street View were not described in detail.
Comment
SPECIFIC OBSERVATIONS
- Line 33 - Suggest to include a discussion in the introduction section about the "so what" of remote sensing road extraction in comparison with traditional cartographic techniques.
Reply
Thank you for your suggestions. We have added these techniques and relevant citations to the Introduction section:
Traditional cartographic techniques are time-consuming and labour-intensive [6,7]. In comparison, remote sensing techniques changed the mapping community fundamentally without relying entirely on surveyed ground measurements [6].
Comment
- Line 208 - When discussing the SAR virtual long antenna and the spatial resolution it is worth mentioning that the long antenna aperture is realised in the cross-range direction and that the range resolution is given by the pulse width (in the general the bandwidth of the signal). Another particularity of the SAR compared with optical sensors is the low correlation between range and spatial resolution.
Reply
Thank you for your kind reminder. We have added them to our manuscript.
Comment
- Line 225 - Change detection is an important technique used in the SAR domain and numerous papers are available which could be referred to here.
Reply
Thank you for your kind reminder. We have added them.
Reviewer 2 Report
The authors did a commendable task of conducting a comprehensive review of active and passive remote sensing for road extraction. The manuscript is nicely written and provide very detailed up-to-date information regarding the data source, techniques etc. used for road extraction using moderate and high resolution satellite images. If it its published, could a great source of literature and add value to the journal readership. In my view, the following things could be considered to improve the article -
- The authors may try to provide some key findings in the literature instead of providing how they structured the manuscript
- Introduction section is comparatively poorly written than other parts; the authors could try to put more focus on the rationale of the study
- The authors could try to provide some specific case example with the accuracy of the findings and limitations and how to improve it
- It would be great if the authors also provide some scopes of future research
Author Response
Comment
The authors did a commendable task of conducting a comprehensive review of active and passive remote sensing for road extraction. The manuscript is nicely written and provide very detailed up-to-date information regarding the data source, techniques etc. used for road extraction using moderate and high resolution satellite images. If it is published, could a great source of literature and add value to the journal readership. In my view, the following things could be considered to improve the article -
Reply
Thank you very much for your positive comments and suggestions. Those comments are all valuable and very helpful for revising and improving our paper and the important guiding significance to our research. We have studied all comments carefully and tried our best to improve the manuscript, and made some changes to the manuscript. These changes will not influence the content and framework of the paper.
Comment
The authors may try to provide some key findings in the literature instead of providing how they structured the manuscript.
Reply
Thank you for your kind reminder. We have added some key findings for the eight tables we summarized. These tables and key findings would help readers compare and understand different kinds of literature. We will also consider your suggestions in our future research.
Comment
Introduction section is comparatively poorly written than other parts; the authors could try to put more focus on the rationale of the study.
Reply
Thank you for your kind reminder. We have revised the Introduction section. We removed some nonsense sentences and put more focus on the rationale of the study.
Comment
The authors could try to provide some specific case example with the accuracy of the findings and limitations and how to improve it.
Reply
Thank you for your kind reminder. In section four, we introduced the combination of multisource data for road extraction, including the combination of high-resolution images with other data, hyperspectral images with other data, SAR images with other data, and LiDAR data with other data. Various examples showed limitations in accuracy using only one data source, and the accuracy was improved with the combination with other data.
Comment
It would be great if the authors also provide some scopes of future research
Reply
Thank you for your kind reminder. We provided it in section 4.5. We revised the subtitle to make it easier to find:
4.5. Some Scopes of Future Research in Road Extraction
Reviewer 3 Report
This paper is an interesting review that synthesizes and compares the results published in a reasonable number of references. Its topic may have practical applications in numerous fields, as well as single or coupled remote sensing data are already used in the new digital cartography. The text could be completed with better images. Therefore, I can think about a wide dissemination of this paper.
Author Response
Comment
This paper is an interesting review that synthesizes and compares the results published in a reasonable number of references. Its topic may have practical applications in numerous fields, as well as single or coupled remote sensing data are already used in the new digital cartography. The text could be completed with better images. Therefore, I can think about a wide dissemination of this paper.
Reply
Thank you very much for your positive comments and suggestions. We added a graphic abstract to help readers study the paper (Please see the attachment). In addition, we summarized eight tables for different techniques to help readers compare and understand different kinds of literature. We also added some key findings for the eight tables we summarized.
